# Spatio-temporal estimates of HIV risk group proportions for adolescent girls and young women across 13 priority countries in sub-Saharan Africa

**Adam Howes**[1,2]*, **Kathryn A. Risher**[2,3], **Van Kính Nguyen**[2,4], **Oliver Stevens**[2], **Katherine M. Jia**[5], **Timothy M. Wolock**[1,2], **Rachel T. Esra**[2], **Lycias Zembe**[6], **Ian Wanyeki**[6], **Mary Mahy**[6], **Clemens Benedikt**[6], **Seth R. Flaxman**[7], **Jeffrey W. Eaton**[2]

**1** Department of Mathematics, Imperial College London, London, United Kingdom, **2** MRC Centre for Global Infectious Disease Analysis, School of Public Health, Imperial College London, London, United Kingdom, **3** Heidelberg Institute for Global Health, Faculty of Medicine, Heidelberg University, Heidelberg, Germany, **4** Department of Public Health Sciences, Penn State College of Medicine, Hershey, PA, United States of America, **5** Center for Communicable Disease Dynamics, Department of Epidemiology, Harvard T.H. Chan School of Public Health, Boston, MA, United States of America, **6** Joint United Nations Programme on HIV/AIDS, Geneva, Switzerland, **7** Department of Computer Science, University of Oxford, Oxford, United Kingdom

* ath19@ic.ac.uk

**Data Availability Statement:** This study involved secondary analysis of household survey data which are publicly accessible upon reasonable request

## Abstract

The Global AIDS Strategy 2021-2026 identifies adolescent girls and young women (AGYW) as a priority population for HIV prevention, and recommends differentiating intervention portfolios geographically based on local HIV incidence and individual risk behaviours. We estimated prevalence of HIV risk behaviours and associated HIV incidence at health district level among AGYW living in 13 countries in sub-Saharan Africa. We analysed 46 geospatially-referenced national household surveys conducted between 1999-2018 across 13 high HIV burden countries in sub-Saharan Africa. Female survey respondents aged 15-29 years were classified into four risk groups (not sexually active, cohabiting, non-regular or multiple partner[s] and female sex workers [FSW]) based on reported sexual behaviour. We used a Bayesian spatio-temporal multinomial regression model to estimate the proportion of AGYW in each risk group stratified by district, year, and five-year age group. Using subnational estimates of HIV prevalence and incidence produced by countries with support from UNAIDS, we estimated new HIV infections in each risk group by district and age group. We then assessed the efficiency of prioritising interventions according to risk group. Data consisted of 274,970 female survey respondents aged 15-29. Among women aged 20-29, cohabiting (63.1%) was more common in eastern Africa than non-regular or multiple partner (s) (21.3%), while in southern countries non-regular or multiple partner(s) (58.9%) were more common than cohabiting (23.4%). Risk group proportions varied substantially across age groups (65.9% of total variation explained), countries (20.9%), and between districts within each country (11.3%), but changed little over time (0.9%). Prioritisation based on behavioural risk, in combination with location- and age-based prioritisation, reduced the proportion of population required to be reached in order to find half of all expected new infections from 19.4% to 10.6%. FSW were 1.3% of the population but 10.6% of all expected new

from the Demographic and Health Survey (DHS) and Population HIV Impact Assessment (PHIA) websites (https://dhsprogram.com/data/available-datasets.cfm; https://phia-data.icap.columbia.edu/datasets). Outputs of our analysis are available from https://github.com/athowes/multi-agyw.

**Funding:** Adam Howes was supported by the EPSRC Centre for Doctoral Training in Modern Statistics and Statistical Machine Learning (EP/S023151/1 to SRF). Adam Howes, Van Kính Nguyen, Kathryn A. Risher, and Jeffrey W. Eaton were supported by the Bill and Melinda Gates Foundation (OPP1190661, OPP1164897 to JWE). Timothy M. Wolock was supported by Imperial College London (President's PhD Scholarship to TMW). Seth R. Flaxman was supported by the EPSRC (EP/V002910/2 to SRF). Jeffrey W. Eaton was supported by UNAIDS and National Institute of Allergy and Infectious Disease of the National Institutes of Health (R01AI136664 to JWE). Lycias Zembe, Ian Wanyeki, Mary Mahy and Clemens Benedikt are employed by UNAIDS. This research was supported by the MRC Centre for Global Infectious Disease Analysis (MR/R015600/1 to JWE), jointly funded by the UK Medical Research Council (MRC) and the UK Foreign, Commonwealth & Development Office (FCDO), under the MRC/FCDO Concordat program and is also part of the EDCTP2 programme supported by the European Union. Lycias Zembe, Ian Wanyeki, Mary Mahy, and Clemens Benedikt are employed by UNAIDS, who also partially funded this study. Lycias Zembe, Ian Wanyeki, Mary Mahy, and Clemens Benedikt contributed to the design of the study, revision of the manuscript, and approved the final manuscript for submission. Findings and conclusions in this manuscript are those of the authors and do not necessarily represent the official position of the funding agencies.

**Competing interests:** I have read the journal's policy and the authors of this manuscript have the following competing interests: Kathryn A. Risher was supported through a consultancy via UNAIDS to conduct an early version of this work; Lycias Zembe, Ian Wanyeki, Mary Mahy, and Clemens Benedikt are employees of UNAIDS; Jeffrey W. Eaton is a member of the editorial board for PLOS Global Public Health; other authors have declared that no competing interests exist.

infections. Our risk group estimates provide data for HIV programmes to set targets and implement differentiated prevention strategies outlined in the Global AIDS Strategy. Successfully implementing this approach would result in more efficiently reaching substantially more of those at risk for infections.

## Introduction

In sub-Saharan Africa, adolescent girls and young women (AGYW) aged 15–29 are 28% of the population but 44% of new HIV infections [1]. HIV incidence amongst AGYW is 2.4 times higher than among similarly aged males, due to structural vulnerability and power imbalances, age patterns of sexual mixing, younger age at first sex, and increased susceptibility to HIV infection [2]. AGYW have therefore been identified as a priority population for HIV primary prevention, with significant investments being made in prevention programming [3, 4].

The Global AIDS Strategy 2021–2026 [5], adopted by the United Nations (UN) General Assembly in June 2021, proposed stratifying packages of HIV prevention provided to AGYW based on both local population-level HIV incidence and individual-level sexual risk behaviour to promote more efficient prioritisation of prevention services [6–9]. Four prioritisation strata (Table A in S2 Text) were defined based on: (1) subnational annual incidence (<0.3%, 0.3–1.0%, 1.0–3.0% and >3.0%), and (2) self-reported high-risk behaviour or recent STI infection. The strategy encourages programmes to define targets for the proportion of AGYW to be reached with a range of interventions (Table B in S2 Text) based upon this prioritisation strata [5]. All AGYW are recommended to have access to a basic package of HIV prevention, while those with with high risk behaviours in moderate incidence settings and all AGYW in very high incidence settings are recommended to have access to enhanced intervention packages [2, 10]. These interventions may include STI screening and treatment, access to pre-exposure prophylaxis (PrEP), access to post-exposure prophylaxis (PEP), comprehensive sexuality education, and economic empowerment.

Implementation of a stratified HIV prevention strategy by national HIV programmes and stakeholders requires data on the population size and HIV incidence in each risk group by location. We developed a multinomial Bayesian spatio-temporal model to estimate the proportion of AGYW aged 15–29 years in four behavioural risk groups, stratified by district, year, and five-year age group. We focused on 13 countries in sub-Saharan Africa which have been identified by the Global Fund to Fight AIDS, TB, and Malaria [4] as priority countries for implementation of AGYW HIV prevention. Our methodology is standardised across countries, allowing prioritisation both within and between countries. We analysed the extent to which the risk group proportions varied across districts, age groups, between countries, and over time. Using our estimates, we calculated the HIV prevalence, people living with HIV (PLHIV), HIV incidence, and expected number of new HIV infections in each risk group by disaggregating district-level estimates from the Naomi model [11]. Finally, we quantified the increased efficiency of HIV prevention, in terms of the expected number of new infections that could be preemptively reached, by stratified prioritisation of interventions by risk group, location and age group.

## Methodology

### Data

We analysed nationally-representative household survey data from 13 countries: Botswana, Cameroon, Kenya, Lesotho, Malawi, Mozambique, Namibia, South Africa, Eswatini, Tanzania,

Uganda, Zambia and Zimbabwe. We included surveys conducted in these countries between 1999 and 2018 in which women were interviewed about their sexual behaviour and sufficient geographic information was available to locate survey clusters to health districts. Demographic and Health Surveys (DHS) [12], AIDS Indicator Surveys (AIS) [13], Population-based HIV Impact Assessment (PHIA) [14] surveys, and the Botswana AIDS Impact Survey 2013 (BAIS) [15] were included.

For each survey, we classified female respondents aged 15–29 years into one of four behavioural risk groups according to reported sexual risk behaviour in the past 12 months. These risk groups were: not sexually active, one cohabiting sexual partner, non-regular or multiple sexual partner(s), and AGYW who report transactional sex (Table 1). In the case of inconsistent responses, women were categorised according to the highest risk group they fell into, ensuring that the categories were mutually exclusive. Exact survey questions varied slightly across survey types and between survey phases (S2 Text). Questions captured information about whether the respondent had been sexually active in the past twelve months, and if so how with many partners. For their three most recent partners, respondents were also asked about the type of partnership (spouse, cohabiting partner, partner not cohabiting with respondent, friend, sex worker, sex work client, and other).

Some surveys included a specific question asking if the respondent had received or given money or gifts for sex in the past twelve months. In these surveys, 2.64% of women reported transactional sex. In surveys without such a question, women almost never (0.01%) answered that one of their three most recent partners was a sex work client. Due to this incomparability across surveys, we did not include surveys without a specific transactional sex question when estimating the proportion of the population who engaged in transactional sex. We focused on estimating the proportion of women who reported transactional sex at a district level, and subsequently adjusted these proportions to align to national estimates for the number of female sex workers.

We used estimates of population, people living with HIV (PLHIV) and new HIV infections stratified by district and age group from HIV estimates published by UNAIDS that were developed using the Naomi model [11]. The model synthesises data from multiple sources to produce subnational estimates of indicators of interest, and has been used by countries as a part of the HIV estimates process supported by UNAIDS. The administrative area hierarchy and geographic boundaries we used correspond to those used for health service planning by countries, exceptions being Cameroon and Kenya where we conducted analysis one level higher at the department and county levels, respectively (Table E in S2 Text). We used the most recent 2022 estimates for all

**Table 1. HIV risk groups and assumed HIV incidence rate ratio for each risk group relative to AGYW with one cohabiting sexual partner.** Among FSW, the incidence rate ratio depended on the level of HIV incidence among the general population. The incidence rate ratio for women with non-regular or multiple sexual partner(s) was derived from analysis of ALPHA network data. Non-regular partners are defined to be non cohabiting. The transactional sex risk group is adjusted during analysis to correspond to female sex worker, and incidence rate ratios among FSW were derived based on patterns of relative HIV prevalence among FSW compared to general population prevalence. When the local HIV incidence in the general population is higher, the incidence rate ratio for FSW is lower.

| Risk group | Description | Local HIV incidence | Incidence ratio |
|---|---|---|---|
| None | Not sexually active | – | 0.0 |
| Low | One cohabiting partner | – | 1.0 (Baseline) |
| High | Non-regular or multiple partner(s) | – | 1.72 |
| | | <0.1% | 25.0 |
| | | 0.1–0.3% | 13.0 |
| Very high | Transactional sex (adjusted to correspond to female sex workers) | 0.3–1.0% | 9.0 |
| | | 1.0–3.0% | 6.0 |
| | | >3.0% | 3.0 |

countries, apart from Mozambique where, due to data accuracy concerns, we used the 2021 estimates (in which the Cabo Delgado province is excluded due to disruption by conflict).

## Two-stage model for sexual risk group proportions

To estimate the proportion of AGYW in each risk group, we took a two-stage modelling approach. First, we fit a spatio-temporal multinomial model stratified by district, year (1999–2018) and five-year age group (15–19, 20–24, and 25–29) for the proportion of AGYW in three categories: (1) not sexually active, (2) one cohabiting partner, and (3) either non-regular or multiple partner(s), or transactional sex. Combining the two highest risk groups (high and very high) in this way allowed data from all surveys to be included in this first stage model. Second, we fit a spatial logistic regression model separating those who have non-regular or multiple partner(s) from those who reported transactional sex, stratified by district, and five-year age group, and using only data from the surveys with a specific transactional sex question. As surveys were only available in the years 2013–2018, we assumed the proportion in the very high risk group among those in the two highest risk groups was constant over time. We combined the two models using 1000 samples from each posterior distribution to produce samples for all four risk groups. Finally, we adjusted the samples from the transactional sex category to match age- and country-specific FSW population size estimates. We modified the samples from the non-regular or multiple partner(s) risk group to ensure that after adjustment the risk group proportions still summed to one. FSW population size estimates by age were obtained by disaggregating national 15–49 FSW population size estimates [16] using the FSW age distribution in South Africa from the Thembisa model [17] in combination with country-specific age at first sex distributions [18]. Further technical details are in Section 1 of S1 Text.

We considered four model specifications for the space-age-time multinomial model for the three risk groups. All models included intercepts for each risk group, as well as age, country, and age-country random effects. To account for district-level variation we used spatial random effects consisting of a parameter for each district. We considered alternative model specifications in which the spatial random effects were either independent or spatially correlated such that more information was shared between neighboring districts than those far apart. Similarly, we used temporal random effects to allow variation in risk group proportions over time, and considered alternative model specifications as independent versus first-order auto-regressive where a smooth temporal trend is assumed. To understand the importance of each part of the model we analysed the relative sizes of the variance parameters for each effect.

For the logistic regression model of the proportion engaging in transactional sex among those with non-regular or multiple partner(s), we considered six specifications. Each included an intercept, age and country random effects, and a spatial random effect allowing district-level variation. Both independent and spatially correlated spatial random effects were considered. To improve estimation with sparse data, during model selection we considered alternatives with national-level covariates for either the proportion of men who reported ever having paid for sex or having paid for sex in the last twelve months [19].

We performed inference using the integrated nested Laplace approximation (INLA) [20] algorithm via the `R-INLA` package [21]. For models with a Gaussian latent field, INLA has comparable accuracy to Markov chain Monte Carlo with realistic, finite samples [22], and is substantially more computationally tractable for high dimensional models like ours, which has 940 districts, 20 years, 3 age groups, and 4 risk groups. It is not possible to directly fit multinomial logistic regression models in `R-INLA`, so we used the multinomial-Poisson transformation [23]. Details of this approach, including how we used Kronecker products over Gaussian Markov random fields to appropriately define random effects, are provided in Section 1.2 of S1 Text.

The best performing model was selected according to the conditional predictive ordinate (CPO) criterion [24], a measure of leave-one-out model performance that can be calculated directly in R-INLA without explicit model refitting [25]. The R [26] code used to implement the models and produce results is available from github.com/athowes/multi-agyw. We used sf [27] for handling of spatial data, orderly [28] for reproducible research, ggplot2 for data visualisation [29] and rticles [30] for reporting via rmarkdown [31].

## HIV indicators and prevention prioritisation

Using risk group proportion estimates, and subnational estimates developed using the Naomi model, we calculated HIV prevalence, PLHIV, HIV incidence and number of new HIV infections stratified by district, age group and risk group. Further details are provided in S1 Text.

**Prevalence and PLHIV.** To calculate HIV prevalence by risk group, we disaggregated the district-age specific prevalence estimates from Naomi estimates to risk groups using odds ratios for the relative odds of having HIV between risk groups, calculated from a logistic regression of country-age specific household survey HIV bio-marker data. Prevalence disaggregation was on the logit scale to ensure that HIV prevalence in each risk group remained in the range 0% to 100%. The number of PLHIV was calculated by multiplying HIV prevalence by risk group population size.

**Incidence and new infections.** We disaggregated HIV incidence by risk group using the HIV infection risk ratios in Table 1. The risk ratio used for the not sexually active risk group was zero, excluding incidence from non-sexual transmission which is negligible in these populations. For the non-regular or multiple partner(s) risk group, the risk ratio was based on analyses of risk factors for incident HIV infection from studies in sub-Saharan Africa [32] and supported by a recent systematic review [9]. Risk ratios for the highest risk group vary based upon general population HIV incidence and are based on an analysis of HIV prevalence among FSW relative to population prevalence [33] using data from the UNAIDS Key Population Atlas [34]. The number of new HIV infections were calculated by multiplying HIV incidence by susceptible risk group population size.

**Infections reached.** For each possible stratification of risk, we calculated the expected number of new infections that would be found per person reached when prioritising according to incidence. To do so, we ordered the strata by descending incidence before cumulatively summing the expected new infections and population. We assumed it was possible to reach all members of every strata.

## Results

### Data

We included 46 surveys in our analysis (Fig 1, Table C in S2 Text), with a total sample size of 274,970 women aged 15–29 years (103,063 aged 15–19 years, 92,173 aged 20–24 years, and 79,734 aged 25–29 years). Of these, 12 surveys included a specific transactional sex question, with a total sample size of 62,853 (28,753 aged 15–19 years, 26,324 aged 20–24 years, and 7,776 aged 25–29 years (There were 6 DHS surveys which excluded women 25–29 from the transactional sex survey question)). The median number of surveys per country was four, ranging from one in Botswana and South Africa to six in Uganda.

### Model selection and model fit

The best fitting multinomial regression model included correlated spatial random effects and independent and identically distributed temporal random effects. The best logistic regression

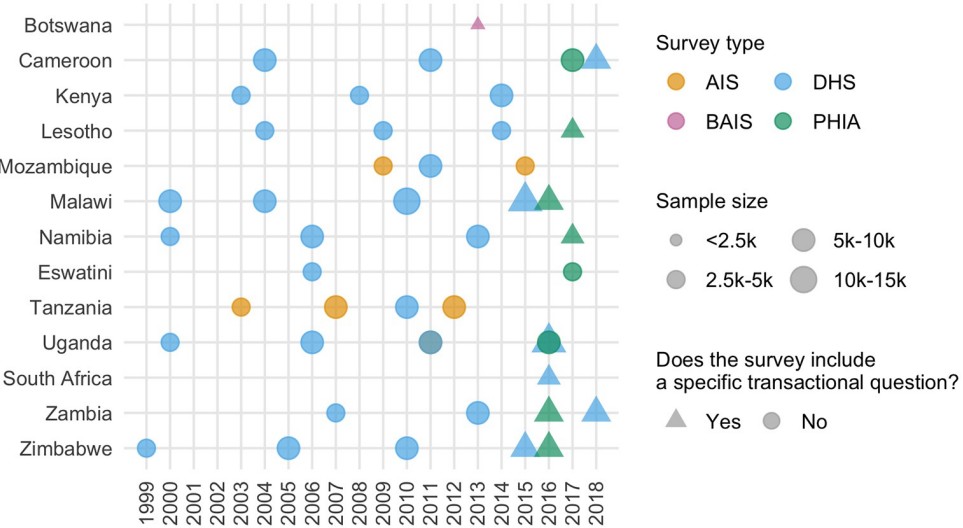

**Fig 1. Surveys used in our analysis by year, survey type, sample size, and whether the survey included a question about transactional sex.** Details of included surveys are in Table C in S2 Text.

model for transactional sex included correlated spatial random effects and the proportion of men who reported ever paying for sex covariate. Model performance according to the CPO criterion is provided in S1 Text for all models we considered. Direct estimates of the four risk group proportions from these surveys were highly correlated with our modelled estimates at a national-level (Figs E—Q in S2 Text).

## Risk group estimates

Figs 2 and 3 show posterior mean estimates for the proportion in each risk group from our final model (provided in S1 Data). In subsequent results, all estimates refer to 2018, the most recent year included in our analysis, unless otherwise indicated.

The median national FSW proportion was 1.1% (95% CI 0.4–1.9) for the 15–19 age group, 1.6% (95% CI 0.6–2.8) for the 20–24 age group and 1.9% (95% CI 0.5–3.5) for the 25–29 age group.

In the 20–24 and 25–29 year age groups, the majority of women were either cohabiting or had non-regular or multiple partner(s). Countries in eastern and central Africa (Cameroon, Kenya, Malawi, Mozambique, Tanzania, Uganda, Zambia and Zimbabwe) had a higher proportion of women in these age groups cohabiting (63.1% [95% CI 35–78.7%] compared with 21.3% [95% CI 10.1–48.8%] with non-regular partner[s]). In contrast, countries in southern Africa (Botswana, Eswatini, Lesotho, Namibia and South Africa) had a higher proportion with non-regular or multiple partner(s) (58.9% [95% CI 43.2–70.5%], compared with 23.4% [95% CI 9.7–39.1%] cohabiting). This clear geographic delineation passes along the border of Mozambique, through the interior of Zimbabwe and along the border of Zambia (Fig 2).

In most districts (57.9%; 95% credible interval [CI] 27.7–79.7) adolescent girls aged 15–19 were not sexually active. The exception was Mozambique, where the majority (64.23%) were sexually active in the past year and close to a third (34.17%) were cohabiting with a partner.

Age group was the most important factor explaining variation in risk group proportions, accounting for 65.9% (95% CI 54.1–74.9%) of total variation. The primary change in risk group proportions by age group occurs between the 15–19 age group and 20–29 age group (Fig 3). The next most important factor was location. Country-level differences explained

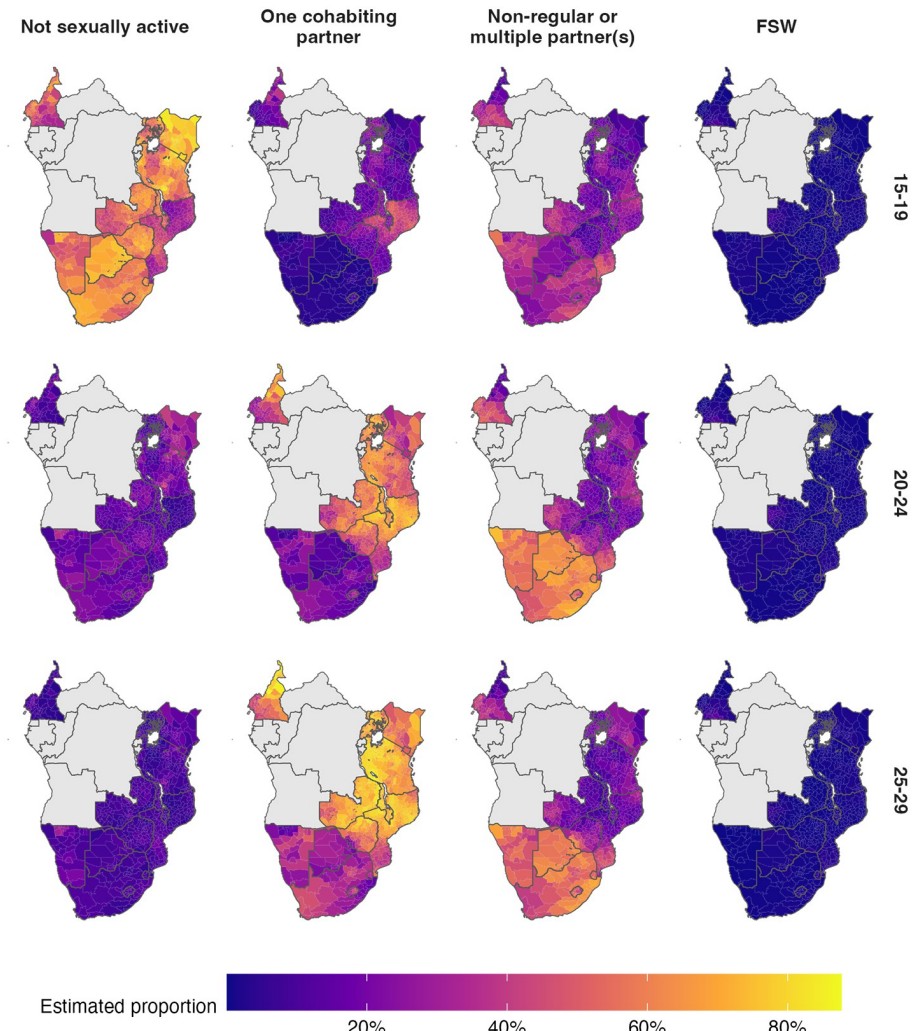

**Fig 2. The spatial distribution (posterior mean) of the AGYW risk group proportions in 2018.** Estimates are stratified by risk group (columns) and five-year age group (rows). Countries in grey were not included in our analysis. District boundaries used under a CC BY license, with permission from UNAIDS, original copyright 2023.

20.9% (95% CI 11.9–34.5%) of variation, while district-level variation within countries explained 11.3% (95% CI 8.2–15.3%). Temporal changes only explained 0.9% (95% CI 0.6–1.4%) of variation, indicating very little change in risk group proportions over time. We observed similar variance decomposition results fitting each country individually, and using other model specifications.

## Expected infections reached

For any given fraction of AGYW prioritised, substantially more new infections were reached by strategies that included behavioural risk stratification. Reaching half of all expected new infections required reaching 19.4% of the population when stratifying by subnational area and age, but only 10.6% when behavioural stratification was included (Fig 4). The majority of this benefit came from reaching FSW, who were 1.3% of the population but 10.6% of all new infections.

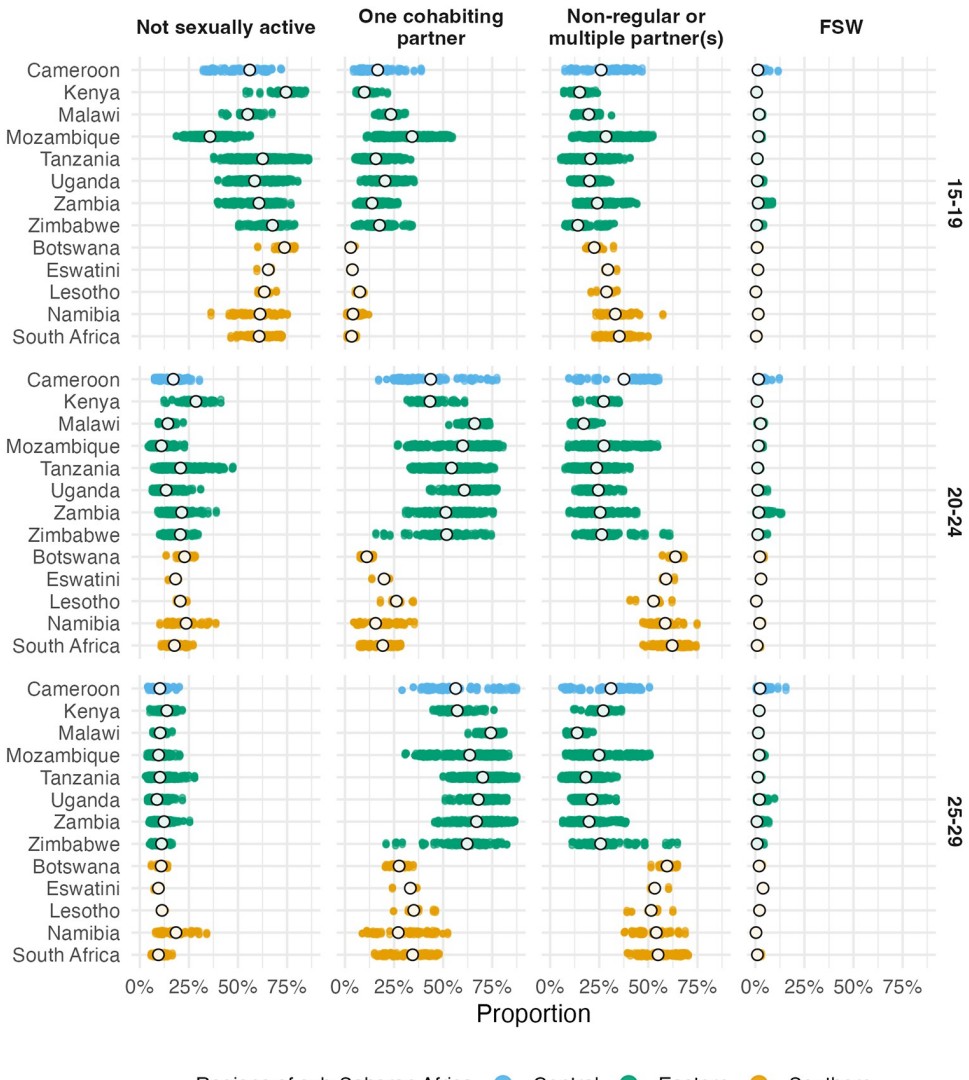

**Fig 3. National (in white) and subnational (in color) posterior means of the risk group proportions.** Estimates are stratified by risk group (columns) and five-year age group (rows).

Considering each country separately, on average, reaching half of new infections in each country required reaching 14.6% (range 8.7–21.8%) of the population when stratifying by area and age, reducing to 5.1% (range 2.1–13.2%) when behaviour was included. The relative importance of stratifying by age, location and behaviour varied between countries, analogous to the varying contribution of each to the total variance (Fig C in S2 Text). For example, FSW in Kenya were estimated to be 1.1% of the population and close to a third (16.1%) of all new infections, whereas FSW in Tanzania were just 1.2% of the population and 16% of all new infections.

## Discussion

We estimated the proportion of AGYW who fall into different risk groups at a district level in 13 sub-Saharan African countries. Our estimates support consideration of differentiated prevention programming according to geographic locations and risk behaviour, as outlined in the

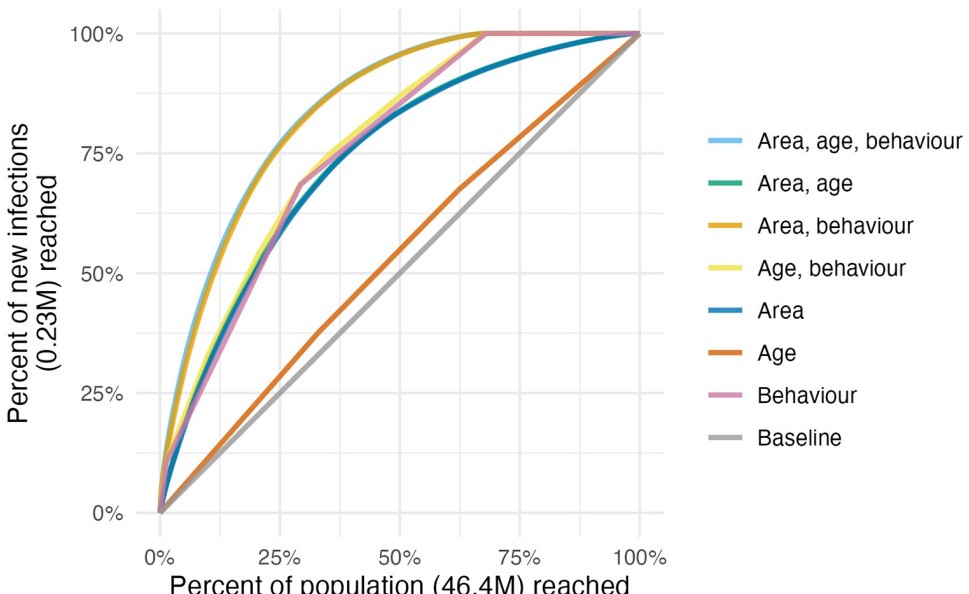

**Fig 4. Percentage of new infections reached across all 13 countries, taking a variety of risk stratification approaches, against the percentage of at risk population required to be reached.**

Global AIDS Strategy. Systematic differences in risk by age groups, and variation within and between countries, explained the large majority of variation in risk group proportions. Changes over time were negligible in the overall variation in risk group proportions. The proportion of 15–19 year olds who are sexually active, and among women aged 20–29 years, norms around cohabitation especially varied across districts and countries. This variation underscores the need for these granular data to implement HIV prevention options aligned to local norms and risk behaviours.

We defined four risk groups based on sexual behaviour, the most proximal determinant of risk. Other factors, such as condom usage or type of sexual act, may account for additional heterogeneity in risk from sexual behaviour. However, we did not include these factors in view of measurement difficulties, concerns about consistency across contexts, and the operational benefits of describing risk parsimoniously. Sexual behaviour confers risk only when AGYW reside in geographic locations where there is unsuppressed viral load among their potential partners.

We did not include more distal determinants, such as school attendance, orphanhood, or gender empowerment, as we expect their effects on risk to largely be mediated by more proximal determinants. However, to effectively implement programming, it is crucial to understand these factors, as well as the broader structural barriers and limits to personal agency faced by AGYW. Importantly, programs must ensure that intervention prioritisation occurs without stigmatising or blaming AGYW.

Brugh et al. [35] previously geographically mapped AGYW HIV risk groups using biomarker and behavioural data from the most recent surveys in Eswatini, Haiti and Mozambique to define and subsequently map risk groups with a range of machine learning techniques. Our work builds on Brugh et al. [35] by including more countries, integrating a greater number of surveys, and connecting risk group proportions with HIV epidemic indicators to help inform programming.

By considering a range of possible risk stratification strategies, we showed that successful implementation of a risk-stratified approach would allow substantially more of those at risk for infections to be identified before infection occurs. A considerable proportion of estimated new infections were among FSW, supporting the case for HIV programming efforts focused on key population groups [36]. There is substantial variation in the importance of prioritisation by age, location and behaviour within each country. This highlights the importance of understanding and tailoring HIV prevention efforts to country-specific contexts. By standardising our analysis across all 13 countries, we showed the additional efficiency benefits of resource allocation between countries.

We found a geographic delineation in the proportion of women cohabiting between southern and eastern Africa, calling attention to a divide attributable to many cultural, social, and economic factors. The delineation does not represent a boundary between predominately Christian and Muslim populations, which is further north. We also note that the high numbers of adolescent girls aged 15–19 cohabiting in Mozambique is markedly different from the other countries [37].

Our modelled estimates of risk group proportions improve upon direct survey results for three reasons. First, by taking a modular modelling approach, we integrated all relevant survey information from multiple years, allowing estimation of the FSW proportion for surveys without a specific transactional sex question. Second, whereas direct estimates exhibit large sampling variability at a district level, we alleviated this issue using spatio-temporal smoothing (Fig B in S2 Text). Third, we provided estimates in all district-years, including those not directly sampled by surveys, allowing estimates to be consistently fed into further analysis and planning pipelines (such as our analysis of risk group specific prevalence and incidence).

The final surveys included in our risk model model were conducted in 2018. We plan to update our analysis with more surveys as they become available, but do not anticipate that the risk group proportions will change substantially, as we found that they did not change significantly over time.

Our analysis focused on females aged 15–29 years, and could be extended to consider optimisation of prevention more broadly, accounting for the 56% of new infections among adults 15–49 which occur in women 30–49 and men 15–49. Estimating sexual risk behaviour in adults 15–49 would be a crucial step toward greater understanding of the dynamics of the HIV epidemic in sub-Saharan Africa, and would allow incidence models to include stratification of individuals by sexual risk.

## Limitations

Our analysis was subject to challenges shared by most approaches to monitoring sexual behaviour in the general population [38]. In particular, under-reporting of higher risk sexual behaviours among AGYW could affect the validity of our risk group proportion estimates. Due to social stigma or disapproval, respondents may be reluctant to report non-marital partners [39, 40] or may bias their reporting of sexual debut [18, 41, 42]. For guidance of resource allocation, differing rates of under-reporting by country, district, year or age group are particularly concerning to the applicability of our results; and, while it may be reasonable to assume a constant rate over space-time, the same cannot be said for age, where aspects of under-reporting have been shown to decline as respondents age [43], suggesting that the elevated risks we found faced by younger women are likely a conservative estimate. If present, these reporting biases will also have distorted the estimates of infection risk ratios and prevalence ratios we used in our analysis, likely over-attributing risk to higher risk groups.

We have the least confidence in our estimates for the FSW risk group. As well as having the smallest sample sizes, our transactional sex estimates do not overcome the difficulties of sampling hard to reach groups. We inherent any limitations of the national FSW estimates [16] which we adjust our estimates of transactional sex to match. Furthermore, we do not consider seasonal migration patterns, which may particularly affect FSW size. More generally, we did not consider covariates potentially predictive of risk group proportions (such as sociodemographic characteristics, education, local economic activity, cultural and religious norms and attitudes), which are typically difficult to measure spatially. Identifying measurable correlates of risk, or particular settings in which time-concentrated HIV risk occurs, is an important area for further research to improve risk prioritisation and precision HIV programme delivery.

The efficiency of each stratified prevention strategy depends on the ability of programmes to identify and effectively reach those in each strata. Our analysis of new infections potentially averted assumed a "best-case" scenario where AGYW of every strata can be reached perfectly, and should therefore be interpreted as illustrating the potentially obtainable benefits rather than benefits which would be obtained from any specific intervention strategy. In practice, stratified prevention strategies are likely to be substantially less efficient than this best-case scenario. Factors we did not consider include the greater administrative burden of more complex strategies, variation in difficulty or feasibility of reaching individuals in each strata, variation in the range or effectiveness of interventions by strata, and changes in strata membership that may occur during the course of a year. Identifying and reaching behavioural strata may be particularly challenging. Empirical evaluations of behavioural risk screening tools have found only moderate discriminatory ability [9], and risk behaviour may change rapidly among young populations, increasing the challenge to effectively deliver appropriately timed prevention packages. This consideration may motivate selecting risk groups based on easily observable attributes, such as attendance of a particular service or facility, rather than sexual behaviour.

## Conclusion

We estimated the proportion of AGYW aged 15–19, 20–24 and 25–29 years in four sexual risk groups at a district-level in 13 priority countries and analyzed the number of infections that could be reached by prioritisation based upon location, age and behaviour. Though subject to limitations, these estimates provide data that national HIV programmes can use to set targets and implement differentiated HIV prevention strategies as outlined in the Global AIDS Strategy. Successfully implementing this approach would result in more efficiently reaching a greater number of those at risk of infection.

Among AGYW, there was systematic variation in sexual behaviour by age and location, but not over time. Age group variation was primarily attributable to age of sexual debut (ages 15–24). Spatial variation was particularly present between those who reported one cohabiting partner versus non-regular or multiple partners. Risk group proportions did not change substantially over time, indicating that norms relating to sexual behaviour are relatively static. These findings underscore the importance of providing effective HIV prevention options tailored to the needs of particular age groups, as well as local norms around sexual partnerships.

## Supporting information

**S1 Text. Mathematical appendix, including for: The risk group model, FSW population size estimation, and calculation of HIV prevalence, HIV incidence and expected new HIV infections reached.**
(PDF)

**S2 Text. Supplementary tables and figures, including for: The Global AIDS strategy, the household survey data used, and plots of country-specific comparisons between direct and modelled estimates, HIV prevalence estimates, HIV incidence estimates, and expected new HIV infections reached.**
(PDF)

**S1 Data. Estimates of risk group proportions.**
(CSV)

## Author Contributions

**Conceptualization:** Kathryn A. Risher, Clemens Benedikt, Jeffrey W. Eaton.

**Data curation:** Adam Howes, Kathryn A. Risher, Van Kính Nguyen, Oliver Stevens, Rachel T. Esra, Ian Wanyeki, Mary Mahy, Jeffrey W. Eaton.

**Formal analysis:** Adam Howes.

**Funding acquisition:** Jeffrey W. Eaton.

**Investigation:** Adam Howes.

**Methodology:** Adam Howes, Kathryn A. Risher, Van Kính Nguyen, Timothy M. Wolock, Seth R. Flaxman, Jeffrey W. Eaton.

**Project administration:** Kathryn A. Risher, Clemens Benedikt, Jeffrey W. Eaton.

**Software:** Adam Howes, Kathryn A. Risher.

**Supervision:** Seth R. Flaxman, Jeffrey W. Eaton.

**Visualization:** Adam Howes.

**Writing – original draft:** Adam Howes.

**Writing – review & editing:** Adam Howes, Kathryn A. Risher, Van Kính Nguyen, Oliver Stevens, Katherine M. Jia, Timothy M. Wolock, Rachel T. Esra, Lycias Zembe, Ian Wanyeki, Mary Mahy, Clemens Benedikt, Seth R. Flaxman, Jeffrey W. Eaton.

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
