## [Decision Letter · Decision Letter 0]

19 Dec 2022

PGPH-D-22-01534

Spatio-temporal estimates of HIV risk group proportions for adolescent girls and young women across 13 priority countries in sub-Saharan Africa

Dear Dr. Howes,

Thank you for submitting your manuscript to PLOS Global Public Health. After careful consideration, we feel that it has merit but does not fully meet PLOS Global Public Health’s publication criteria as it currently stands. Therefore, we invite you to submit a revised version of the manuscript that addresses the points raised during the review process.

The reviewers have raised a number of questions regarding your analysis; please ensure you respond to each of these questions when revising your manuscript.

We look forward to receiving your revised manuscript.

Kind regards,

Hugh Cowley

Staff Editor

Journal Requirements:

1. Please send a completed 'Competing Interests' statement, including any COIs declared by your co-authors. If you have no competing interests to declare, please state "The authors have declared that no competing interests exist". Otherwise please declare all competing interests beginning with the statement "I have read the journal's policy and the authors of this manuscript have the following competing interests:"

a. State what role the funders took in the study. If the funders had no role in your study, please state: “The funders had no role in study design, data collection and analysis, decision to publish, or preparation of the manuscript.”

b. If any authors received a salary from any of your funders, please state which authors and which funders.

3. Please ensure that the funders and grant numbers match between the Financial Disclosure field and the Funding Information tab in your submission form. Note that the funders must be provided in the same order in both places as well.

4. Please provide separate figure files in .tif or .eps format only and remove any figures embedded in your manuscript file. Please also ensure that all files are under our size limit of 10MB.

5. Figure 2: please (a) provide a direct link to the base layer of the map (i.e., the country or region border shape) and ensure this is also included in the figure legend; and (b) provide a link to the terms of use / license information for the base layer image or shapefile. We cannot publish proprietary or copyrighted maps (e.g. Google Maps, Mapquest) and the terms of use for your map base layer must be compatible with our CC-BY 4.0 license. 

Additional Editor Comments (if provided):

Reviewers' comments:

Reviewer's Responses to Questions

**Comments to the Author**

1. Does this manuscript meet PLOS Global Public Health’s publication criteria? Is the manuscript technically sound, and do the data support the conclusions? The manuscript must describe methodologically and ethically rigorous research with conclusions that are appropriately drawn based on the data presented.

Reviewer #1: Yes

Reviewer #2: Yes

2. Has the statistical analysis been performed appropriately and rigorously?

Reviewer #1: Yes

Reviewer #2: Yes

3. Have the authors made all data underlying the findings in their manuscript fully available (please refer to the Data Availability Statement at the start of the manuscript PDF file)?

Reviewer #1: Yes

Reviewer #2: Yes

4. Is the manuscript presented in an intelligible fashion and written in standard English?

Reviewer #1: Yes

Reviewer #2: Yes

5. Review Comments to the Author

Reviewer #1: In the manuscript, explain why you use the INLA not WINBUGS coding?

The multinomial regression could be modeled in WINBUGS directly.

Is there no other potential covariate that could be used for better modeling?

Please explain sub-national effect more clearly. Why and how you used it?

why the interaction term for spatiotemporal effect didn't consider in the modeling framework?

Reviewer #2: This is a well-crafted manuscript investigating the spatio-temporal estimates of HIV risk group proportions for adolescent girls and young women across 13 priority countries in sub-Saharan Africa. Their analyses identify specific age groups at the district level that should be targeted for HIV intervention in SSA. This is critical in reducing the HIV epidemic in the southern region of SSA. In addition to the main figures, the supplementary Tables and Figures show country-by-country risk, mostly among female sex workers for all age groups. With the help of their models, specific resources can target at-risk populations with a moderate assurance of how many people to reach and where these resources should go.

My little concern is about using different data from UNAIDS Key Population Atlas apart from the DHS, which is the may source data for the analyses. I believe the two variants of data are based on different designs, and combining them may not result in dependable results. It would have been more attainable if the UNAIDS data had been used in their sensitivity analysis to confirm the results from the DHS data.

Figure 1 is not clear. I recommend that the authors use a table as an alternative visualization.

6. PLOS authors have the option to publish the peer review history of their article (what does this mean?). If published, this will include your full peer review and any attached files.

**Do you want your identity to be public for this peer review?** For information about this choice, including consent withdrawal, please see our Privacy Policy.

Reviewer #1: No

Reviewer #2: **Yes: **Ayodeji E. Iyanda

---

## [Editor Report · Decision Letter 1]

23 Feb 2023

Spatio-temporal estimates of HIV risk group proportions for adolescent girls and young women across 13 priority countries in sub-Saharan Africa

PGPH-D-22-01534R1

Dear Mr Howes,

We are pleased to inform you that your manuscript 'Spatio-temporal estimates of HIV risk group proportions for adolescent girls and young women across 13 priority countries in sub-Saharan Africa' has been provisionally accepted for publication in PLOS Global Public Health.

Best regards,

Lucy Chimoyi, PhD

Academic Editor